Feasibility of using chest computed tomography (CT) imaging at the first lumbar vertebra (L1) level to assess skeletal muscle mass: a retrospective study

http://orcid.org/0000-0003-1448-3827 Liu Shaohua 1 2
Han Xia 3
Li Jianjun 4
Xie Xia 5
Yang Yunkai 6
Jiang Wangyan 1 2
Liu Li 1 2 liliu@tjh.tjmu.edu.cn
Liu Zhelong 1 2 liuzhelong@163.com
1 Department of Endocrinology, Tongji Hospital, Tongji Medical College, Huazhong University of Science and Technology , Wuhan , China
2 Branch of National Clinical Research Center for Metabolic Diseases , Wuhan, Hubei , China
3 Wuhan Wuchang Hospital , Wuhan , China
4 Department of Radiology, Tongji Hospital, Tongji Medical College; Huazhong University of Science and Technology , Wuhan , China
5 School of Computer Science and Technology, Hainan University , Haikou , China
6 Eight-year Program of Clinical Medicine, Tongji Hospital, Tongji Medical College, Huazhong University of Science and Technology , Wuhan , China
Barak Meir
Electronic publication date: 2023 Dec 11
Publication date: 2023
Volume: 11
Electronic Location ID: e16652
Received 2023 Jun 21; Accepted 2023 Nov 20
Copyright: © 2023 Liu et al.
Copyright year: 2023
Copyright holder: Liu et al.
License: This is an open access article distributed under the terms of the Creative Commons Attribution License, which permits unrestricted use, distribution, reproduction and adaptation in any medium and for any purpose provided that it is properly attributed. For attribution, the original author(s), title, publication source (PeerJ) and either DOI or URL of the article must be cited.
License URL: https://creativecommons.org/licenses/by/4.0/

Keywords: Sarcopenia, Skeletal muscle mass, Computed tomography, The first lumbar vertebra

Funding: Teaching Research Fund of Huazhong University of Science and Technology 2021144 and 2022141 Sen-Mei China Diabetes Research Fund Z-2017-26-1902 This work was supported by the Teaching Research Fund of Huazhong University of Science and Technology (No. 2021144 to Zhelong Liu and No. 2022141 to Li Liu) and the Sen-Mei China Diabetes Research Fund (No. Z-2017-26-1902). The funders had no role in study design, data collection and analysis, decision to publish, or preparation of the manuscript.

==============================
Background

Skeletal muscle mass is an essential parameter for diagnosing sarcopenia. The gold standard for assessing skeletal muscle mass is using computed tomography (CT) to measure skeletal muscle area at the third lumbar vertebra (L3) level. This study aims to investigate whether skeletal muscle mass could be evaluated at the first lumbar vertebra (L1) level using images obtained from routine chest CT scans.

Methods

Skeletal muscle index (SMI, cm2/m2) and skeletal muscle density (SMD, HU) are commonly used to measure relative muscle mass and the degree of fat infiltration. This study used CT images at the L1 level to measure the skeletal muscle area (SMA, cm2) in 815 subjects from the health examination center. Linear regression analysis was used to explore the association between L1 and L3 measurements. The receiver operating characteristic (ROC) analysis was used to assess the predictive performance of L1 SMI for sarcopenia. The sex-specific cut-off values for low skeletal muscle mass in patients under the age of 60 were determined using the following formula: “mean − 1.28 × standard deviation.” A multivariate linear regression model was established.

Results

A significantly higher SMI at the L1 level was found in males than in females (43.88 ± 6.33 cm2/m2 vs 33.68 ± 5.03 cm2/m2; P < 0.001). There were strong correlations between measures at the L1 and L3 levels in both the total subject and sex-specific analyses. A negative association was found between age and L3 SMI in males (r = −0.231, P = 0.038). Both body mass index (BMI) and body surface area (BSA) were positively associated with L1 SMI in both males and females. A multivariate analysis was used to establish a prediction rule to predict SMI at the L3 level. The assessment of consistency and interchangeability between predicted and actual SMI at the L3 level yielded moderately good results. Considering the significant differences observed between male and female participants, the sex-specific cut-off values of the L1 SMI for defining low skeletal muscle mass were 36.52 cm2/m2 in males and 27.29 cm2/m2 in females.

Conclusions

Based on a population from central China, the correlated indicators obtained at the L1 level from routine chest CT scans may serve as effective surrogate markers for those at the L3 level in assessing overall skeletal muscle mass.

Introduction

Sarcopenia, first proposed by Rosenberg et al. in 1989 (Rosenberg, 1997), is a geriatric syndrome characterized by age-related loss of skeletal muscle mass, muscle strength and/or degraded physical performance (Chen et al., 2014). Sarcopenia may increase the risk of falls, fractures, frailty, dyskinesia, poor quality of life, longer hospital stays and even death (Cruz-Jentoft et al., 2019). Sarcopenia also reduces surgical survival (Dolan et al., 2019) and impacts the prognosis of various diseases including cardiac disease (Kang et al., 2019), respiratory disease (Hirai et al., 2019), diabetes (Cheng et al., 2017; Mori, Kuroda & Matsuhisa, 2019), liver disease (Ooi et al., 2019) and cancer (Deng et al., 2018). With the worldwide population aging, the prevention and treatment of sarcopenia is an urgent need. However, there is still inadequate understanding of the condition, leading to a lack of uniform diagnostic methods and standards.

The Asian Working Group for Sarcopenia (AWGS) defines sarcopenia as low muscle mass plus low muscle strength and/or low physical performance (Chen et al., 2014). To diagnose sarcopenia, seven diagnostic criteria have been proposed by AWGS, with reduced skeletal muscle mass being a required parameter in all criteria (Akishita et al., 2018). Current methods for measuring skeletal muscle mass include bioelectric impedance analysis (BIA), dual X-ray absorptiometry (DXA), computed tomography (CT), and magnetic resonance imaging (MRI). CT and MRI are considered the current gold standards for the assessment of skeletal muscle mass (Chen et al., 2014), but the high cost and time requirements and infrequent routine use of MRI sharply limit the use of MRI measurements in most clinical settings. CT scans, however, are widely used in disease diagnosis, preoperative evaluation, and follow-up exams as they can distinguish between muscle tissue, adipose tissue, and internal organs (Guerri et al., 2018). Previous research has shown that CT measurements of cross-sectional SMA in specific body parts accurately reflect the muscle mass in those areas (Mitsiopoulos et al., 1998). Although cross-sectional SMA at the L3 level is highly correlated with whole-body skeletal muscle mass (Hamaguchi et al., 2016; Nishikawa et al., 2016; van der Werf et al., 2018), the L3 level cut-off value for sarcopenia requires abdominal CT imaging and cannot be measured with chest CT scans, which are more commonly used. Some previous studies have found that SMA at the L1 level shows a strong correlation with SMA at the L3 level and with total skeletal muscle mass (Derstine et al., 2018; Kim et al., 2016). As L1 level CT images can be easily obtained during chest CT scans without additional radiation or cost, evaluating muscle mass using L1 level CT images could significantly expand the routine assessment of low skeletal muscle mass.

SMI and SMD are crucial parameters for measuring sarcopenia and myosteatosis, respectively, and are both related to physical function (Rollins et al., 2020; Williams et al., 2017). SMI is calculated by normalizing the SMA for height: SMA (cm2)/height2 (m2) (Baumgartner et al., 1998). Due to the influence of various factors on SMI (Anderson, Liu & Garcia, 2017; Hai et al., 2017), a uniform cutoff value has not yet been established.

The objective of this study was to explore the association of skeletal muscle mass measurements obtained from L1 and L3 level CT images and to assess the feasibility of using chest CT images at the L1 level to evaluate skeletal muscle mass.

Materials and Methods

This study was a single-center retrospective cross-sectional study conducted in the Tongji Hospital of Tongji Medical College of Huazhong University of Science and Technology.

Study population

A total of 886 individuals who underwent comprehensive general health examinations (including chest CT scans) at Tongji Hospital Health Examination Center were initially included in the study. Individuals aged below 20 or above 80 years, those with neurological or wasting diseases affecting muscle strength, severe gastrointestinal conditions or eating disorders, significant movement disorders, cardiopulmonary dysfunction, and those without cross-sectional images at the L1 level were excluded from the study. After applying these exclusion criteria, 71 participants were excluded: nine individuals not meeting the age criteria, 19 with cardiopulmonary abnormalities, five with impaired motor function, seven with severe gastrointestinal disorders or eating disorders, and 31 without scans at the L1 level. The final study cohort was comprised of 815 generally healthy participants, including 131 subjects who had CT scans that included both L1 and L3 levels. Patient information on sex, age at the time of CT scan, height, body weight and ethnicity was obtained from the Health Examination Center’s registration form. This study was approved by the Ethics Committee of Tongji Hospital of Tongji Medical College of Huazhong University of Science and Technology (TJ-IRB20191201) and conducted in accordance with the Declaration of Helsinki. Because existing CT scans were used retrospectively, the requirement for informed consent was waived.

CT scans evaluation

Non-contrast CT scans with a slice thickness of 0.5 cm were used from the entire chest CT scans, with additional CT scanning parameters, including a 64-row detector configuration, 0.5 sec/rotation, and a tube voltage of 120 kV. All CT images were obtained from the Image Archiving and Communication System (PACS), and were anonymized and viewed utilizing Sante DICOM Viewer software. A professional imaging physician selected the transverse images at the L1 level and the L3 level. The L1 and L3 segments were identified starting with the first thoracic vertebra. If the scan did not encompass the first thoracic vertebra, the 12th thoracic and sacral joints were used to identify the L1 and L3 segments.

Parameters measurement

The SMA of the selected CT images were measured by Slice-Omatic software V5.0 (Tomovision, Magog, Quebec, Canada), using Hounsfield unit (HU) ranges for muscle tissue of −29 to +150 HU (Recio-Boiles et al., 2018; Steele et al., 2021). The skeletal muscles at the L1 level included abdominal wall muscles, intercostal muscles, diaphragm, psoas muscles and paraspinal muscles (Kang et al., 2019). The muscles at the L3 level were composed of psoas, paraspinal and abdominal wall muscles. An example of L1 and L3 delineation using Slice-Omatic is shown in Fig. 1, where the red areas represent skeletal muscles.

Figure 1 Examples of L1 and L3 delineation using Slice-Omatic software.

Cross-sectional CT images at the L1 (1A) and L3 (1B) levels with skeletal muscles marked in red. After selecting the region of interest for skeletal muscle, the SMD was displayed automatically. Abbreviations: L1, the first lumbar vertebra; L3, the third lumbar vertebra; SMA, skeletal muscle area.

Statistical analysis

The normal distribution of continuous variables was assessed using the Kolmogorov-Smirnov test. The data with a normal distribution and continuous variables were presented as means ± standard deviation (SD), and the difference in means between the two groups was tested using a student’s t-test. Abnormally distributed continuous data were expressed as medians (interquartile range), and the comparison was done using the Mann-Whitney U test. Correlation tests were performed using Pearson correlation coefficients and linear regression. Categorical variables were expressed as percentage and numbers, and were compared using the Chi-square test. The SMI at the L3 level was used as the standard for classifying subjects with low skeletal muscle mass. Previous studies have established cutoff values for diagnosing low skeletal muscle mass using L3 SMI as 44.7 (cm2/m2) in males and 33.0 (cm2/m2) in females (Derstine et al., 2018; van der Werf et al., 2018; Zeng et al., 2021). Using the SMI values obtained from actual measurements at the L3 level as the diagnostic criteria, an ROC analysis was conducted to assess the predictive efficacy of muscle mass at the L1 level for sarcopenia. The cutoff values for L1 SMI in subjects younger than 60 years were then calculated using the “mean − 1.28 × SD” method. Kappa analysis was used to verify its consistency with the L3 diagnostic method. A multiple linear regression model was also established. Given the significant influence of age, gender, and weight on muscle mass, these clinical variables were introduced into the regression model to further explore the association between the L3 SMI and L1 SMI. The predictive formula obtained from the linear regression analysis was used to predict SMI at the L3 level. To visualize the agreement between the actual measured values and predicted values of the L3 SMI, a Bland-Altman plot was constructed. In this plot, the Y-axis represents the differences between the two sets of values, while the X-axis represents the means of those values. Deming regression and Passing-Bablok regression analyses were performed to assess the compatibility and potential interchangeability between the actual measured L3 SMI and the predicted values. Body surface area (BSA) was calculated using the Dubois formula. In line with the methods described in a previous retrospective article (Li et al., 2023), we utilized SPSS version 20.0 software for the statistical analysis of all data and GraphPad Prism (ver. 9, GraphPad Software; La Jolla, San Diego, CA, USA) for plotting. We considered statistical significance as a two-sided P-value of less than 0.05.

Results

Baseline characteristics of study population

The study population comprised of 532 males and 283 females who were healthy and recruited from the physical examination center of Tongji hospital. The median age was 52 years (range, 22–79 y) for male participants and 51 years (range, 23–79 y) for female participants. The height, weight, BMI, BSA, SMA and SMI of male subjects were all significantly higher than those of female subjects. Detailed information about the total study population, both male and female, is presented in Table 1.

Table 1 The basic characteristics of the 815 study subjects.

Characteristics	Overall population	Male subjects	Female subjects	P-value	
Age (year old)	51 (45, 58)	52 (46, 58)	51 (43, 58)	0.051	
Height (m)	1.67 (1.61, 1.73)	1.71 (1.67, 1.74)	1.59 (1.56, 1.63)	<0.001	
Weight (kg)	67.5 (58.4, 75.6)	72.2 ± 9.6	56.5 (51.8, 62.1)	<0.001	
BMI (kg/m2)	24.1 ± 3.0	24.8 ± 2.8	22.1 (20.5, 24.4)	<0.001	
BSA (m2)	1.84 (1.70, 1.97)	1.92 ± 0.14	1.67 (1.59, 1.75)	<0.001	
SMA (cm2)	113.00 ± 26.55	127.59 ± 19.65	85.59 ± 12.31	<0.001	
L1 SMI (cm2/m2)	40.34 ± 7.65	43.88 ± 6.33	33.68 ± 5.03	<0.001	
Notes:

Data are presented as mean ± SE or median (interquartile range). P-values comparing male subjects and female subjects are from Student’s t-test or Mann–Whitney U-test.

BMI, body mass index; BSA, body surface area; SMA, skeletal muscle area; L1, first lumbar vertebra; SMI, skeletal muscle index.

Association between L1 SMI and sex, age, BMI and BSA

Age was negatively correlated with L3 SMI among male subjects (r = −0.231, P = 0.038; Fig. 2A), however, no statistically significant association was observed with L1 SMI (P = 0.387). In female subjects, no significant association was found between age and either L1 or L3 SMI (L1: P = 0.253, L3: P = 0.755; Fig. 2B). A significantly higher SMI was observed in males than in females (P < 0.001, Fig. 2C). SMI was significantly lower in subjects aged ≥60 y than in those <60 y at both the L1 and L3 levels (L1: P = 0.001, L3: P = 0.024; Fig. 2D). There was a significant positive correlation between BMI and L1 SMI both in males (r = 0.676, P < 0.001) and females (r = 0.553, P < 0.001; Fig. 3A), and a positive correlation was found between BSA and L1 SMI in both males (r = 0.416, P < 0.001) and females (r = 0.220, P < 0.001; Fig. 3B). L3 SMI was also positively correlated with BMI in both male (r = 0.589, P < 0.001) and female (r = 0.566, P < 0.001) subjects (Fig. 3C), and with BSA in both male and female subjects (male subjects: r = 0.338, P < 0.001; female subjects: r = 0.289, P = 0.042; Fig. 3D).

Figure 2 The association between SMI and gender/age.

(A) The association between SMI and age in male subjects. (B) The association between SMI and age in female subjects. (C) Sex differences in SMI. (D) The comparison of SMI between young (20–59 years) and old (≥60 years) subjects. Abbreviations: L1, the first lumbar vertebra; L3, the third lumbar vertebra; SMI, skeletal muscle index.

Figure 3 The association between SMI and BMI/BSA.

(A) The correlation between BMI and the L1 SMI. (B) The correlation between BSA and L1 SMI. (C) The correlation between BMI and L3 SMI. (D) The correlation between BSA and L3 SMI. Abbreviations: BMI, body mass index; BSA, body surface area; L1, the first lumbar vertebra; L3, the third lumbar vertebra; SMI, skeletal muscle index.

Correlation analysis between L1 and L3 muscle mass measures

Out of the 815 total subjects, only 131 had CT images at both the L1 and L3 levels, so the correlation analysis of L1 and L3 indicators was performed exclusively within this subset of 131 subjects. An analysis of disparities in the population characteristics between these 131 subjects and the broader cohort of 815 subjects was performed. There were no significant differences between the subset and the total study population in terms of sex, age and L1 SMA, indicating that the sample was representative (Table 2).

Table 2 Analysis of differences in population characteristics between 131 subjects and the 815 total subjects.

	Sex (Male: Female)	Age (year old, M (P25, P75))	L1 SMA (cm2, x¯±s)	
131 subjects	81:50	52 (46, 63)	108.12 ± 27.24	
Overall subjects	532:283	51 (45, 58)	113.00 ± 26.55	
χ2/Z/t	0.578	−1.551	1.922	
P	0.444	0.121	0.055	
Notes:

Data are presented as mean ± SE or median (interquartile range). P-values comparing 131 subjects and total study population are from Student’s t-test, Mann–Whitney U-test or Chi-square test.

L1, first lumbar vertebra; SMA, skeletal muscle area.

A correlative analysis and linear regression analysis were conducted to investigate the association between L1 and L3 related measures in the 131 subjects. The results showed significant positive correlations between L1 level and L3 level CT images in SMI (r = 0.802, P < 0.001) and SMA (r = 0.866, P < 0.001) in both male and female subjects (SMI: male subjects: r = 0.767, P < 0.001; female subjects: r = 0.625, P < 0.001. SMA: male subjects: r = 0.770, P < 0.001; female subjects: r = 0.609, P < 0.001. Figure S1 in the online supplementary section). Additionally, SMD at L1 and L3 showed a strong correlation (r = 0.903, P < 0.001), which was consistent in both male and female subjects in a subgroup analysis (male subjects: r = 0.879, P < 0.001; female subjects: r = 0.912, P < 0.001), as shown in Fig. 4. There was a modest correlation between SMD and SMA at the L3 level.

Figure 4 The association between L1 SMD and L3 SMD and the association between SMD and SMA.

(A) The correlation between L1 SMD and L3 SMD in all subjects. (B) The correlation in men and women, respectively. (C) The correlation between SMD and SMA at the L1 level. (D) The correlation between SMD and SMA at the L3 level. Abbreviations: L1, the first lumbar vertebra; L3, the third lumbar vertebra; SMD, skeletal muscle density.

As presented in Table 3, the following prediction rules were established: L3 SMI = 0.956 × L1 SMI + 8.142. A multivariate model was also established and is displayed in Table 3. Age, sex, and weight were all significant covariates, and L1 SMI remained the strongest predictor of L3 SMI. The new multifactor prediction rule was established, as follows: L3 SMI = 0.682 × L1 SMI −0.078 × Age + 0.105 × weight + 5.011 × Sex + 12.943 (use value ‘‘0” for female sex and ‘‘1” for male sex). The linear regression analysis results suggest that the measured value of L3 SMI is highly correlated with the predicted value (r = 0.869, P < 0.001; Fig. 5A). Next, the Bland-Altman plot was used to compare the actual and predicted SMI at the L3 level (Fig. 5B), which showed moderately good agreement (ICC = 0.861). Figure 6 illustrates the results of the Deming regression (Fig. 6A), which show the formula for the relationship between the actual measured L3 SMI and the predicted values: Predicted L3 SMI = 6.622 + 0.894 × L3 SMI (with a 95% CI for the intercept: [3.107–10.15]; and a 95% CI for the slope: [0.764–0.933]). The Passing-Bablok regression (Fig. 6B) results provided the following formula: Predicted L3 SMI = 5.083 + 0.885 × L3 SMI (with a 95% CI for the intercept: [1.424–8.490]; and a 95% CI for the slope: [0.807–0.970]). Both of these regressions indicate a moderate consistency and interchangeability between the predicted values of L3 SMI and the actual measured values.

Table 3 Regression analysis.

	Covariates	Coefficient	r (95% CI)	P-Value	
Basic prediction rule	L1 SMI	0.956	0.801 [0.813–1.080]	<0.001	
Prediction rule:	L3 SMI = 0.956 × L1 SMI + 8.142	
Backward-selected multivariate mode	L1 SMI	0.682	0.656 [0.544–0.821]	<0.001	
	Age	−0.078	−0.238 [−0.134 to −0.022]	0.007	
	Weight	0.105	0.176 [0.001–0.209]	0.047	
	Sex	5.011	0.437 [3.191–6.831]	<0.001	
Prediction rule:	L3 SMI = 0.682 × L1 SMI − 0.078 × Age + 0.105 × weight + 5.011 × Sex + 12.943	
Notes:

When using the multivariate prediction rule, use value ‘‘0” for female sex and ‘‘1” for male sex.

L1, first lumbar vertebra; L3, third lumbar vertebra; SMI, skeletal muscle index.

Figure 5 The association between actual SMI and predicted SMI at the L3 level.

(A) Linear regression analysis. (B) Bland-Altman plots showing the consistency between both methods. Difference (diff) between measurements is shown on the Y-axis and average value of the two measurements is shown on the X-axis. Abbreviations: L3, the third lumbar vertebra; SMA, skeletal muscle area.

Figure 6 The assessment of potential interchangeability between the actual measured L3 SMI and predicted values.

(A) Deming regression; (B) Passing-Bablok regression.

The sex-specific cut-off values of L1 SMI for defining low skeletal muscle mass

An ROC analysis was employed to assess the predictive performance of muscle mass at the L1 level for L3-defined sarcopenia. As shown in Fig. 7, the AUC value was 0.8626 (95% CI = [0.7684–0.9567], P < 0.001) in males and 0.8300 (95% CI [0.7098–0.9502], P = 0.001) in females. Based on: mean − 1.28 × SD among subjects aged <60 years, the cut-off value for the L1 SMI for the diagnosis of sarcopenia was 36.52 cm2/m2 in male subjects and 27.29 cm2/m2 in female subjects. Kappa analysis was used to test the consistency between the diagnosis method at the L1 level and that at the L3 level, and the obtained kappa value was 0.475, indicating moderate agreement.

Figure 7 ROC analysis of L1 SMI for the prediction of L3-defined low skeletal muscle mass.

(A) Among male subjects; (B) among female subjects. The ROC analysis was performed on the subset of 131 patients who had CT scans that included both L1 and L3 levels.

Discussion

Sarcopenia is an important factor in the diagnosis of malnutrition and is a risk factor for adverse clinical outcomes in many diseases (Kang et al., 2018; Mitsiopoulos et al., 1998; Rollins et al., 2020). Early screening, early diagnosis and timely treatment are of great significance in improving clinical outcomes. Using L1 SMI measurements is a new approach to assessing low muscle mass and has not been commonly used in previous studies on sarcopenia. To the best of our knowledge, this is the first study to investigate the feasibility of L1 SMI in assessing muscle mass in a Chinese population. This study demonstrated a high correlation between L1 and L3 level measures (SMI, SMD), and the same results were obtained in subgroup analyses of male and female subjects, suggesting that routine chest CT scans may substitute for abdominal CT scans in assessing whole-body muscle mass. This study also provides a reference cut-off value for L1 SMI in a population from central China, giving clinicians and researchers a screening tool to improve the identification rate of sarcopenia patients without additional radiation exposure, patient burden or cost.

A growing body of recent research has used CT scans to evaluate skeletal muscle mass, mainly in study populations of cancer patients (Kim et al., 2016; Nipp et al., 2018; Ooi et al., 2019), surgery patients (Chan & Chok, 2019; Dolan et al., 2019), patients in intensive care units (Toptas et al., 2018) or patients with common chronic diseases such as cardiovascular disease, diabetes, liver disease or digestive system disease (Cheng et al., 2017; Hirai et al., 2019; Kang et al., 2019; Mori, Kuroda & Matsuhisa, 2019). Sarcopenia incidence is increasing and many factors can increase the risk of its occurrence and development. These factors interact with each other and are closely related, making it difficult to create a unified standard of CT diagnosis of sarcopenia.

Previous studies have explored the use of L1 SMI to assess low muscle mass and have attempted to determine cut-off values (Hirai et al., 2019; Kang et al., 2019; Recio-Boiles et al., 2018). A study in Korea (Kang et al., 2019) used cross-sectional CT images at the L1 level to evaluate skeletal muscle mass and investigate the impact of low skeletal muscle mass on the prognosis of 475 coronary artery disease patients who underwent successful percutaneous coronary intervention. The study demonstrated a sex-specific ROC curve for predicting all-cause mortality outcomes using L1 SMI and found optimal sex-specific critical values for low SMI of 31.00 cm2/m2 for males and 25.00 cm2/m2 for females. Another study was conducted in the same year on a healthy US population, where skeletal muscle mass was measured using cross-sectional CT images at each vertebral level between the tenth thoracic (T10) and fifth lumbar (L5) vertebra (Derstine et al., 2018). The study found that the cut-off values of the L1 SMI were 34.60 cm2/m2 in men and 25.90 cm2/m2 in women, which are lower than the values obtained in our study. A study conducted in the United States (US) on 37 patients with non-small cell lung cancer demonstrated that L1 SMI could be used to evaluate musculoskeletal mass in these patients using routine chest CT scans. However, in this study, SMI was normalized for BSA (Recio-Boiles et al., 2018). The differences between these cutoff values could be due to variations in the underlying health conditions and different racial backgrounds of the study participants. These differences may also be influenced by the study population characteristics, measuring tools for SMA, study outcomes and methods of determining the cut-off value. Further research may be needed to investigate specific cutoff values within particular racial or disease-specific populations.

Skeletal muscle mass differs between males and females. In our study, the SMI in men was 1.30-fold higher than in women, which is consistent with previous studies (van der Werf et al., 2018). Skeletal muscle mass generally decreases with age, and it has been suggested that the rate of decline is essentially the same across genders and races (Shaw, Dennison & Cooper, 2017). In van der Werf et al. (2018), a significant negative linear correlation was found between age and L3 SMI in both males and females among 420 healthy Caucasian individuals. A similar correlation was also observed between age and SMI in 541 adult donors for living donor liver transplantation in Japan (Hamaguchi et al., 2016), particularly in subjects aged ≥50 years. The SMI in subjects aged <50 years was 1.20-fold higher than that ≥50 years in both sexes. However, in our study, a gradual decline in L1 SMI with aging was only observed in male subjects, and this decline became significant around the age of 60. In contrast, no significant change in SMI with age was found in female subjects, which differs from previous research findings. This could be attributed to various factors. Changes in sex hormone levels play an essential role in age-related skeletal muscle wasting. Gender differences in muscle wasting have been reported (Anderson, Liu & Garcia, 2017), and the incidence of sarcopenia is lower in females than in males. The declines in muscle mass, muscle strength, and physical function also vary between males and females due to their distinct hormonal profiles (Anderson, Liu & Garcia, 2017). Although some clinical trials of estrogen replacement therapy in postmenopausal women found estrogen replacement therapy helped women maintain muscle strength and function, there is insufficient evidence to support its effect on muscle mass, particularly in well-functioning older women (Sipilä et al., 2013; Taaffe et al., 2005). These findings suggest that age-related changes in female hormones may not have a significant effect on muscle mass. However, it has been established that testosterone has anabolic effects on skeletal muscle. Several studies have shown a slight but significant correlation between testosterone levels and muscle mass (Anderson, Liu & Garcia, 2017), and more pronounced muscle wasting in older men may be attributed to a sharp decline in testosterone levels (Mitchell et al., 2012). As a result, women tend to experience less overall and age-related muscle loss than men (Churchward-Venne, Breen & Phillips, 2014). The lack of significant age-related SMI changes in women in this study may also be because older women in China may have participated in heavy physical labor during their younger years due to the lack of mechanized agricultural production at that time. Given that independent physical activity is known to be beneficial for maintaining skeletal muscle mass (Gao et al., 2015), older Chinese women may have been able to effectively preserve their muscle mass. Additionally, although age-related muscle wasting remains a key consideration in diagnosing sarcopenia, according to the AWGS 2019 criteria (Chen et al., 2020), numerous studies have shown that loss of muscle mass in older adults is influenced not only by aging but also by factors such as diseases, risk factors, personal conditions and lifestyle. An individual’s physical condition during young adulthood, which can be influenced by factors like low birth weight, prepubertal and pubertal growth and other factors, may also impact muscle mass in older adults (Cruz-Jentoft & Sayer, 2019; Shaw, Dennison & Cooper, 2017). Therefore, sarcopenia is increasingly recognized as a condition associated with multiple causal factors (Cruz-Jentoft & Sayer, 2019).

Cut-off values for low muscle mass may vary depending on the measurement method used, the normalized method of SMI and the study population. Most screening methods lack clear cut-off values and further validation in sample populations of different ethnicities is required to determine appropriate cut-off values. Determining a cut-off value for sarcopenia, while convenient for determining prevalence in a particular population, also may not be sufficient to assess individual risk, especially for patients whose measurements are close to the cut-off value. Advances in equipment and software have made it possible to accurately measure muscle content in older adults. Older adults should be screened for low muscle mass and poor functional ability to promote early detection and treatment of sarcopenia.

This study has several limitations that should be acknowledged. This study is a single-center study conducted at Tongji hospital, which is located in the central region of China. Due to differences in living environment, lifestyles, and eating habits in various areas, the characteristics of the study population are not entirely uniform, and the study sample size is limited. Because of this, the application of the cut-off value obtained in our study is somewhat limited. The study population also consisted solely of individuals from a medical examination center. Further verification is required to determine whether this cut-off value can be generalized to the general population or to patients with chronic diseases. Finally, the measurement of musculoskeletal area in this study was a semi-automatic measurement assisted by software, and the scanning parameters could impact the measurement results. Consequently, there may be inherent measurement bias. Considering these limitations, future research in the field of sarcopenia should address these concerns and investigate the clinical implications and validity of the obtained cut-off values in a more comprehensive manner.

Conclusions

Skeletal muscle mass is a crucial factor in the diagnosis of sarcopenia, and CT images at the L1 level may provide a feasible method for assessing skeletal muscle mass. Using conventional chest CT imaging as an alternative to abdominal CT imaging can significantly improve clinical efficiency. In this study, the reference values for L1 SMI were determined to be <38.22 (cm2/m2) in males and <31.55 (cm2/m2) in females. However, it is important to note that skeletal muscle mass is influenced by various factors including age, race, comorbidities and body weight. Therefore, further research is necessary to establish specific cut-off values for L1 levels, taking into account the characteristics of different populations, such as diverse ethnic groups and individuals with varying comorbidities. Adjustments and validations are needed to ensure the accuracy and applicability of these cut-off values in specific contexts.

Supplemental Information

Supplemental Information 1 Overall population.

Click here for additional data file.

Supplemental Information 2 131 study populations with both L1 and L3 data.

Click here for additional data file.

Supplemental Information 3 The association between L1 SMI/SMA and L3 SMI/SMA.

A: The association between L1 SMI and L3 SMI. B: The association in men and women, respectively. C: The association between L1 SMA and L3 SMA.D: The association in men and women, respectively. Abbreviations: L1, the first lumbar vertebra; L3, the third lumbar vertebra; SMI, skeletal muscle index.

Click here for additional data file.

The authors thank the staff at the Department of Endocrinology and Medical records, Tongji Hospital, Tongji Medical College, Huazhong University of Science and Technology, and all the patients who participated in the study.

Additional Information and Declarations

Competing Interests

Author Contributions

Human Ethics

Data Availability

The authors declare that they have no competing interests.

Shaohua Liu performed the experiments, analyzed the data, prepared figures and/or tables, authored or reviewed drafts of the article, and approved the final draft.

Xia Han performed the experiments, analyzed the data, prepared figures and/or tables, authored or reviewed drafts of the article, and approved the final draft.

Jianjun Li analyzed the data, prepared figures and/or tables, and approved the final draft.

Xia Xie analyzed the data, authored or reviewed drafts of the article, and approved the final draft.

Yunkai Yang analyzed the data, authored or reviewed drafts of the article, and approved the final draft.

Wangyan Jiang analyzed the data, authored or reviewed drafts of the article, and approved the final draft.

Li Liu conceived and designed the experiments, prepared figures and/or tables, and approved the final draft.

Zhelong Liu conceived and designed the experiments, authored or reviewed drafts of the article, and approved the final draft.

The following information was supplied relating to ethical approvals (i.e., approving body and any reference numbers):

Ethics Committee of Tongji Hospital of Tongji Medical College of Huazhong University of Science and Technology (Ethical Application Ref: TJ-IRB20191201).

The following information was supplied regarding data availability:

The raw measurements are available in the Supplemental Files.

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
