# Peer review of "Feasibility of using chest computed tomography (CT) imaging at the first lumbar vertebra (L1) level to assess skeletal muscle mass: a retrospective study"

_PeerJ, doi:10.7717/peerj.16652_

## Round 0.1 · original submission · Major Revisions

Dear Dr. Han,

Your manuscript titled "Feasibility of using chest computed tomography (CT) imaging at the first lumbar vertebra (L1) level to assess skeletal muscle mass" was considered by two expert reviewers and based on their opinions and my review, The decision is “major revision”.

Please carefully read the reviewers’ comments and address them fully in your revised manuscript. In addition, please address the following points:

1. Please note that since the study was done only on a Chinese population, the conclusions should be addressed only to the same population (as the authors indicated in line 210 and lines 241-8). Additional studies should support the correlation between L1 and L3 indexes in other populations (e.g., Caucasian). The authors need to incorporate a similar statement in their manuscript (e.g., L177-8, abstract’s conclusions section), and consider removing the second aim from their manuscript (L89-93, as reviewer #1 suggested).

2. Please pay close attention to reviewer #2 comment about your possible overestimation of L1 SMI cut-off values. This point must be addressed and explained.

3. Please address reviewer #1 concerns regarding the use of accurate professional English language and the correct placement of information in the materials and methods and results sections. Also, please be clearer about your study population (L98: if these are healthy individuals, why did they undergo chest CT scans?), and add the number of excluded subjects (total n = 71) per criterion.

4. Statistical analysis – please specify what test was used to determine normal distribution (L129).

5. Some minor edits:
- Line 131: “Abnormally distributed continuous data were expressed as M…”. Please explain.
- Lines 154-5: Figure 2A and 2B shows the relationship between SMI and age for both L1 and L3, yet the results only discuss L1. Please edit the text.
- Line 169 and 183 – typo, space missing in “L1SMA”
- Line 199: typo, power values should be in superscript.

- Figure 5: Please correct y-axis labels for fig. 5C and 5D.
- Figure 6: Please add units to x- and y-axes labels in Fig. 6A. Additionally, what are the units for the Y-axis in figure 6B? I don’t think it is (cm^2/m^2).
- Table 3: please revise table. Multiple spaces are missing between numbers, parentheses, and commas.

**Language Note:** The Academic Editor has identified that the English language must be improved. PeerJ can provide language editing services - please contact us at copyediting@peerj.com for pricing (be sure to provide your manuscript number and title). Alternatively, you should make your own arrangements to improve the language quality and provide details in your response letter. – PeerJ Staff

Reviewer 1 ·

Basic reporting

Please improve the use of professional English language to ensure that the audience will clearly understand the study and its results, and pay extra attention to consistent use of terminology throughout the manuscript (e.g. connection, correlation, association..). The methods and results section should be more demarcated as some comments in the results section belong to the methods section (and vice versa). Please be objective and do not interpret in the results section.

Experimental design

The authors have conducted a retrospective cohort study in 815 healthy individuals. Their objective was to explore the correlation of skeletal muscle mass measurements obtained from L1 and L3 level CT-scans to assess the feasibility of utilizing chest CT-scans at L1 level to evaluate skeletal muscle mass. They also investigated the association (in the manuscript called 'relationship') between L1 SMI and sex, age BMI and BSA to provide guidance for diagnosing and screening for sarcopenia.

The origin of the study population is not very clear. Patients underwent chest CT-scans in a medical examination center, and some disorders have been indicated as exclusion criteria. I am now assuming that this is a healthy population, as no specific disease characteristics of patients are described.

The finding that skeletal muscle mass measured at L1 correlates well with skeletal muscle mass measured at L3 is interesting, but only tells us that Method A is associated with method B. The correlation does not reflect the interchangeability of both methods, whilst the next step in this study is to assess the feasibility of utilizing chest CT-scans at L1. I would suggest to take an extra step in betwen and study whether the assocation between methods is linear, and whether methods can be compared by performing e.g. deming regression/passing ba-blok regression.

Sex-specific cut-off values are established by using L3 cut-offs from earlier studies (Derstine et al. 2018 and van der Werf et al. 2018). However, these previous studies were performed in Caucasian cohorts, while the study population of the current study is an (assumingly healthy) Asian cohort. The authors of the current study do describe the difference between ethnic groups as a weakness of the current study in the discussion section. However, skeletal muscle mass differs greatly between etnicities and it is very likely that such cut-offs cannot just be extrapolated to other ethnic groups. The establishment of cut-off values in the current study is therefore methodologically incorrect, and I suggest to remove this aim from the manuscript.

Validity of the findings

See earlier comments

Additional comments

- Please provide CC and exact p-values in your results section
- For the association between L1 and L3 SMA; this is not significant, but p-value is 0.055, can you elaborate on this?
- Please move Figure 4 to the supplements as this finding is to be expected
- The authors should be aware of the fact that they are studing an association in this retrospective cohort study, and not an association (or even connection), which is interchangeably used in the manuscript
- Figure 4A does not show what is stated in the results section
- Inclusion criteria were age >20 years and <80 years, but Figures also show younger- and older patients
- When performing a t-test, you test for differences; not comparisons

Reviewer 2 ·

Basic reporting

## BASIC REPORTING
Sarcopenia is geriatric syndrome characterized by an age-related loss of skeletal muscle mass and muscle strength, closely associated with wide range of multi-system comorbidities. Despite its significance in medical care, a uniform diagnostic measurement for sarcopenia has not established so far. In the present study conducted by Xia H et al., the authors have investigated the diagnostic criteria for CT-determined sarcopenia at the L1-vertebral level (38.22 cm2/m2 in man, 31.55 cm2/m2 in female) in Chinese subjects undergoing routine medical examinations. Skeletal muscle measurement at the L1 vertebral level was conducted using the definition empolyed in the previous study (DO Kang et al., J Clin Med 2019), as a composite of abdominal wall muscles, intercostal muscles, diaphragm, psoas muscles, and paraspinal muscles. The manuscript is well prepared and sufficient background information has been presented in the introduction section. The overall quality of article's structure, figure, tables and raw data seems to be acceptable.

Experimental design

## EXPERIMENTAL DESIGN
The main research question, which aims to establish a new reference cut-off value of L1 SMI for determining low skeletal muscle mass in the Chinese population, is well defined. The present study's results are expected to fill the current knowledge gap and contribute to the global evidence for diagnosing sarcopenia. The analytic method appears to be adequately described with sufficient details.

Validity of the findings

## VALIDITY OF THE FINDINGS
The key finding of this study is presenting a new reference cut-off value of L1 SMI to determine low skeletal muscle mass. However, there are some points that need clarification:

Major)
1) The authors mentioned using the cut-off value of L3 level as the reference standard in the ROC analysis. However, it remains uncler which L3 SMI value was used to determine the presence of low skeletal muscle mass, estimated L3 SMI (overall population, n=886) or acutally measured L3 SMI (L1-L3 matched population, n=131)? This point requires clear clarification.

2) The Kappa statitics comparing L1 versus L3 SMI to determine low skeletal muscle mass showed moderate agreement ((k=0.57; 0.40~0.59 generally regarded as moderate agreement). The authors should consider toning down their expression of "medium to high consistency".

3) Results: "These results indicate that relevant indexes at the L1 level can replace those at the L3 level as effective indexes to evaluate the overall skeletal muscle mass and fat infiltration."
--> While L1 measurement can be considered an effective surrogate for L3 and overall skeletal muscle mass, complete replacement cannot be claimed solely based on the findings of the present study. This point should be moderated accordingly.

4) As noted by the authors, the suggested reference cut-off value for determining low skeletal muscle mass appears to be somewhat higher compared to previous studies of L1 SMI (Derstine BA. Sci Rep 2018. DO Kang. J Clin Med 2019). Given the potential ethnic differences, claiming a higher cut-off value in the Chinese population, in comparison to that of the US population (Derstine BA et al. Sci Rep 2018), might pose substantial challenges and require further investigation. Therefore, it is crucial for the authors to consider the suggested cut-off value as provisional and provide additional reference cut-off value, such as "mean - 2SD," based on the current expert consensus (Cruz-Jentoft, AJ et al. EWGSOP consensus. Age Ageing 2010). While acknowledging that the current study population included not exclusively young subjects, adopting this alternative method seems appropriate since the majority of participants were healthy individuals.

Minor issues)
1) Results: "Linear regression analysis was 170 conducted to investigate the relationship between L1 and L3 related measures in 131 subjects." --> It appears that authors have conducted both correlative analysis (Figure 4 and Figure 5) and linear regression (Table 3) in this study. This sentence should be corrected appropriately.

2) Results: "As shown in Figure 4A, we can establish the following prediction rules: L3 SMI= 0.956×L1
180 SMI + 8.142." --> The annotation indicating the result (not provided in Figure 4A) should be corrected to Table 3.

3) Results: "There was a relatively weak but statistically significant correlation between SMD and SMA at corresponding levels. --> This should be corrected to "modest correlation" rather than "weak".

4) Results: ICC value should be provided in Figure 6B concerning interobserver variability.

Additional comments

None

---

## Round 0.2 · Minor Revisions

Dear Dr. Han,

Your manuscript titled " Feasibility of using chest computed tomography (CT) imaging at the first lumbar vertebra (L1) level to assess skeletal muscle mass " was reconsidered by an expert reviewer and based on their opinion and my review, the decision is “minor revision”.

Please carefully read the reviewer’s comments and address them fully in your revised manuscript. In addition, please address the following points:

1. Figure 4 was moved to an “online supplementary” section, but the figure itself is missing a title and legend. Currently the image is attached as an “x.eps” file. Please add the supplementary figure to a word/pdf document and add title and legend. In addition, the figure is referred to in the main text (L192) as Fig. S1, please add a clearer reference (e.g., Fig. S1 in the online supplementary section).


Please ensure that all review, editorial, and staff comments are addressed in a response letter and any edits or clarifications mentioned in the letter are also inserted into the revised manuscript where appropriate.

Please note that submitting a revision of your manuscript does not guarantee eventual acceptance, and that your revision may be subject to re-review by the reviewer(s) before a decision is rendered.

Reviewer 2 ·

Basic reporting

The authors have adequately addressed most of the major concerns raised by the reviewer. Overall quality of the manuscript has been improved substantially after revision. I would like to suggest further clarification regarding the ROC curve analysis (Figure 7).

1) According to your reply (Reviewer #2, point 1), my understanding is that the ROC analysis of L1 SMI (Figure 7) was conducted by applying the L3 diagnostic criteria to define low skeletal muscle mass in the subset of 131 patients who had actually measured L3 SMI values. Given that the manuscript subsequently introduces the L1 SMI diagnostic criteria of "mean - 1.28xSD", the authors should provide more explicit details in the title and legend of Figure 7 regarding the variables used to prevent confusion.

2) The followings are my suggestion for improved clarification.
- Figure Title: (e.g.) Figure 7. ROC analysis of L1 SMI for the prediction of L3-defined low skeletal muscle mass
- Figure Legend: Additional clarification is needed to specify that this ROC analysis was performed on the subset of 131 patients, not on the entire population.
- Main manuscript (Line 217-218): (e.g.) An ROC analysis was employed to assess the predictive performance of muscle mass at the L1 level for L3-defined sarcopenia.

Experimental design

No further issue to raised. The study design has been improved substantially after revision.

Validity of the findings

Additional clarification concerning the ROC curve analysis (Figure 7) is necessary. Please see section "1. Basic Reporting" for details.

Additional comments

I have no further comments to be added at this time.

---

## Round 0.3 · accepted · Accept

Dear Dr. Han,

Thank you for submitting your revised manuscript titled " Feasibility of using chest computed tomography (CT) imaging at the first lumbar vertebra (L1) level to assess skeletal muscle mass: A retrospective study ". After reading the revised manuscript I’m happy to let you know that decision is “accept”.